# Acupuncture Modulates Intracranial Self-Stimulation of the Medial Forebrain Bundle in Rats

**DOI:** 10.3390/ijms22147519

**Published:** 2021-07-14

**Authors:** Seong Shoon Yoon, Jaesuk Yun, Bong Hyo Lee, Hee Young Kim, Chae Ha Yang

**Affiliations:** 1Department of Physiology, College of Korean Medicine, Daegu Haany University, Daegu 42158, Korea; hykim@dhu.ac.kr; 2College of Pharmacy and Medical Research Center, Chungbuk National University, Osongsaengmyeong 1-ro, Osong-eup, Heungdeok-gu, Cheongju, Chungbuk 28160, Korea; jyun@chungbuk.ac.kr; 3Department of Acupuncture, Moxibustion and Acupoint, College of Korean Medicine, Daegu Haany University, Daegu 42158, Korea; dlqhdgy@dhu.ac.kr

**Keywords:** acupuncture, intracranial self-stimulation, rats, medial forebrain bundle, cocaine, brain reward function

## Abstract

Acupuncture affects the central nervous system via the regulation of neurotransmitter transmission. We previously showed that Shemen (HT7) acupoint stimulation decreased cocaine-induced dopamine release in the nucleus accumbens. Here, we used the intracranial self-stimulation (ICSS) paradigm to evaluate whether HT stimulation regulates the brain reward function of rats. We found that HT stimulation triggered a rightward shift of the frequency–rate curve and elevated the ICSS thresholds. However, HT7 stimulation did not affect the threshold-lowering effects produced by cocaine. These results indicate that HT7 points only effectively regulates the ICSS thresholds of the medial forebrain bundle in drug-naïve rats.

## 1. Introduction

Psychostimulant use is increasing rapidly among young people in Asia, Europe and the United States [1,2,3]. Although many efforts have been made to prevent psychostimulant abuse, no relevant pharmacotherapy has been approved by the Food and Drug Administration. Intracranial self-stimulation (ICSS) is a behavioral procedure that quantitatively assesses the brain reward function of laboratory animals [4,5,6]. Electrical stimulation of the medial forebrain bundle (MFB) indirectly activates the mesolimbic dopamine (DA) pathways [6]. Cocaine reliably facilitates ICSS of the MFB [7,8] and is often interpreted as an abuse-related effect [6]. Such cocaine-induced facilitation of ICSS appears to involve activation of the mesolimbic DA system [9,10,11]. For example, ICSS promotes DA transmission in the nucleus accumbens (NAc), as do abused drugs that increase extracellular DA release [9,11]. In contrast, ICSS behavior is attenuated dose-dependently by drugs that decrease DA levels or inactivate DA receptors [12]. These results suggest that enhanced ICSS is mediated (to some extent) via stimulation of the mesolimbic DA system.

Recently, acupuncture treatment of drug addiction has become of interest worldwide [13,14,15]. We previously showed that Shemen (HT7) acupoint stimulation decreased acute cocaine-induced DA release and attenuated cocaine-induced sensitization of extracellular DA levels in the NAc via activation of gamma-amino-butyric acid (GABA) neurons in the ventral tegmental area (VTA) [16]. Additionally, methamphetamine-induced DA enhancement in the NAc was reduced by acupuncture at HT7 [17]. These results suggest that such acupuncture could control the mesolimbic DA system by regulating DA release. Thus, we hypothesized that acupuncture, which suppresses cocaine-induced mesolimbic DA neuronal activity, would diminish the reinforcing effects of MFB ICSS. We thus used the ICSS paradigm to explore whether HT7 acupuncture modulates the brain reward function of rats. In the first experiment, we determined whether the reinforcing effects of electrical stimulation of the MFB were modulated by HT7 stimulation. In the second experiment, we examined the effects of acupuncture on the ICSS-facilitating effect of cocaine. To the best of our knowledge, this is the first attempt to investigate the effects of acupuncture on the brain reward function using an ICSS paradigm in an animal model.

## 2. Results

Table 1 shows baseline mean value from each group for ICSS reward thresholds and M_50_ values during baseline sessions before acupuncture treatment. These baseline measures did not significantly differ across group.

### 2.1. Effects of Acupuncture on ICSS in Drug-Naïve Rats

Figure 1 shows the averaged data from the two 15 min test components (30 min of testing) after acupuncture of drug-naïve rats. One- way ANOVA revealed a significant main effect of acupuncture treatment on the ICSS threshold (F(2,17) = 3.624; *p* < 0.05) and M_50_ value (F(2,17) = 5.266; *p* < 0.05). Post-hoc analysis revealed a significant difference in the ICSS thresholds between the HT7 and control groups (*p* < 0.05, Fisher’s LSD). However, the ICSS thresholds did not differ significantly between the SP6 and control groups (*p* = 0.6145, Fisher’s LSD). Similarly, rats undergoing HT7, but not SP6, stimulation showed significant increases in M_50_ values compared to the control group. The M_50_ values differed significantly between the control and HT7 groups (*p* < 0.001, Fisher’s LSD). Figure 2 shows averaged data from the 1 h test sessions after the acupuncture of drug-naïve rats. In contrast to the 30 min test results, HT7 acupuncture slightly increased the ICSS threshold compared to that of the control group. No significant difference in the ICSS thresholds or M_50_ values was observed between the HT7 and control groups. Figure 2D,E shows time-dependent effects of acupuncture on ICSS thresholds and maximum rates, respectively. Two-way ANOVA with repeated measures revealed that HT7 acupuncture had a time-dependent effect on ICSS thresholds (Group × Time interaction; *F*(6,51) = 2.897, *p* < 0.05) and maximal rates (Group × Time interaction; F(6,51) = 0.8250, *p* = 0.5560). Post-hoc analysis revealed a significant difference in the ICSS thresholds between the HT7 and control groups at 15 min after acupuncture stimulation (*p* < 0.01, Fisher’s LSD). Similarly, the ICSS thresholds differed significantly between the HT7 and SP6 groups at 15 min (*p* < 0.05, Fisher’s LSD). Maximum rates did not differ significantly between the HT7 and control groups.

### 2.2. Effects of Acupuncture on the Cocaine-Induced Shifts in the Frequency–Rate Curve

Cocaine (3.2 mg/kg) facilitated ICSS and reduced the ICSS thresholds and M_50_ values. One-way ANOVA revealed a significant main effect of the treatment on the ICSS threshold (F(3,25) = 9.324; *p* < 0.001) and M_50_ value (F(3,25) = 7.340; *p* < 0.05). The control group exhibited significantly lower ICSS reward thresholds (*p* < 0.01, Fisher’s LSD) and M_50_ values (*p* < 0.05, Fisher’s LSD) than did the saline-treated group (Figure 3B,C). However, HT7 acupuncture did not alter the threshold-lowering effects of cocaine. The ICSS thresholds differed significantly between the HT7 and SP6 groups (*p* < 0.05, Fisher’s LSD). Similarly, the M50 values also differed significantly between the HT7 and SP6 groups (*p* < 0.01, Fisher’s LSD). Figure 3D,E shows time (pass)-dependent effects of combined treatment with acupuncture and cocaine on ICSS thresholds and maximum rates, respectively. Two-way ANOVA with repeated measures revealed that cocaine had a time-dependent effect on ICSS thresholds (Group × Time interaction; F(9,75) = 3.674, *p* < 0.05) and maximal rates (Group x Time interaction; F(9,75) = 3.303, *p* < 0.01). Post-hoc analysis revealed a significant difference in the ICSS thresholds between the control group and saline groups at 15 min (*p* < 0.01, Fisher’s LSD) and at 30 min (*p* < 0.05, Fisher’s LSD) after acupuncture. However, there was not a significant difference in the ICSS thresholds between the control group and HT7 groups at any time point. Similarly, post-hoc analysis revealed a significant difference in the maximum rates between the control group and saline groups at 15 min (*p* < 0.01, Fisher’s LSD) and at 30 min (*p* < 0.05, Fisher’s LSD) after acupuncture. However, there was not a significant difference in the maximum rates between the control group and HT7 groups at any time point.

## 3. Discussion

We found that acupuncture at HT7 significantly increased the ICSS thresholds, as revealed by the rightward shifts of the frequency–rate curves of drug-naïve rats compared to controls (see Figure 1). Similarly, rats receiving HT7 stimulation showed significant increases in M_50_ values compared to the control group. These results suggest that stimulation of HT7 points affected the brain reward functions of drug-naïve rats, which are hypothesized to reflect attenuation of the acute rewarding effects of electrical stimulation. Earlier work showed that when the stimulating electrode was located in the MFB or VTA, ICSS per se promoted DA release in the NAc [18], and this was enhanced by drugs of abuse that intrinsically increase extracellular DA levels in the NAc [11,19]. Additionally, ICSS responsiveness was blocked by drugs that deplete DA or block DA receptors [10,20,21,22]. Thus, direct activation of MFB neurons is sufficient to maintain the ICSS. Although the precise effects of acupuncture on the ICSS thresholds remain unclear, a possible explanation is that HT7 stimulation may elevate the brain threshold by regulating GABA_B_ receptors. This hypothesis is supported by our previous finding that acupuncture suppressed reinstatement of the cocaine reward via the GABA receptor system, and enhanced extracellular GABA overflow in the VTA of cocaine-naïve rats [16]. The novel GABA_B_-positive modulator GS39873 or the GABA_B_ receptor agonist baclofen modulated the ICSS thresholds [23]. For example, a higher dose of baclofen significantly elevated the ICSS thresholds of drug-naïve rats and GS39873 or baclofen attenuated facilitation of the cocaine reward effects [23]. Together, the data suggest that acupuncture elevates ICSS thresholds via the GABA_B_ receptor system. However, some limitations to the present study should be noted. We lack direct evidence that acupuncture enhances the threshold or M_50_ effect via GABA_B_ receptor regulation. Further work is needed to clarify the precise mechanisms. Figure 2 shows averaged data from the 1 h test sessions after the acupuncture of drug-naïve rats. No significant difference in the ICSS thresholds or M_50_ values was observed between the HT7 and control groups. These results would seem to indicate that HT7 acupuncture exerts a relatively short-term effect on the reward function of the brain.

In the second experiment, cocaine facilitated ICSS response and reduced the ICSS thresholds and M_50_ values (Figure 3). These data parallel other findings; cocaine enhanced the brain stimulation reward [7,24,25,26]. In contrast, HT7 acupuncture did not alter the threshold-lowering effects of cocaine, indicating that HT7 stimulation does not affect the reward-facilitating effects of cocaine. This result is inconsistent with previous findings. We earlier showed that acupuncture at HT7 reduced cocaine challenge-induced DA release in the NAc and suppressed cocaine-seeking behavior via the activation of VTA GABA neurons [16]. Perhaps relevantly, in the present work, the threshold-enhancing effects of acupuncture were observed immediately after application, as shown in the first experiment, and the acupuncture effect persisted for only 30 min. Thus, one possible explanation of the discrepancy is that the threshold-lowering effects of cocaine are not affected by acupuncture because any effect is relatively short-term. It is also possible that HT7 acupuncture could not surmount the reward-facilitating effects of cocaine and the acute rewarding effects of MFB electrical stimulation. Several reports have shown that extracellular DA release in the NAc is enhanced by both cocaine and ICSS per se. Thus, HT7 stimulation may not suppress the enhanced extracellular DA overflow to the NAc. Additionally, we used only a single dose of cocaine (3.2 mg/kg) in the present study. Thus, we cannot rule out the possibility that HT7 acupuncture might be effective in blocking threshold-lowering effects at a lower dose of cocaine (e.g., 1.0 mg/kg). However, because no significant effects were observed at the 1.0-mg dose of cocaine (data not shown), we could not confirm any effect of acupuncture at this dose. Further work is necessary to elucidate the precise mechanism of the acupuncture effect.

In conclusion, we found that specific HT7 acupoint, but not SP6 acupoint, stimulation significantly increased the ICSS reward threshold, but HT acupuncture did not affect the reward-facilitating effects induced by cocaine. Acupuncture at HT7 only regulated the MFB ICSS thresholds of drug-naïve rats.

## 4. Materials and Methods

### 4.1. Animals

Male Sprague-Dawley rats (Daehan Animal, Seoul, Korea) weighing 325–350 g at the beginning of the experiment were housed individually in a temperature (23 ± 3 °C)- and humidity (30–70%)-controlled animal room under a 12/12 h light/dark cycle (800–2000 h). Water and food were freely available, except during the behavioral ICSS experiments. Six male rats were used in the control and HT7 groups, and seven males were used in the Sanyinjiao (SP6) group in the first experiment. In the second experiment, the rats were divided into four groups: the saline (7 males), control (8 males), HT7 (7 males), and SP6 (7 males) groups.

### 4.2. Intracranial Self-Stimulation (ICSS)

#### 4.2.1. Apparatus

The ICSS experiment was performed in an operant testing chamber (Med Associates, St. Albans, VT, USA). Each operant chamber contained a metal wheel manipulandum (2 × 5 cm) that protruded 1.8 cm from one wall, three stimulus lights (red, yellow, and green) centered 7.6 cm above the wheel, and a 2-W house light. Electrodes were connected to the stimulator via bipolar cables routed through a swivel commutator (Model SL2C; Plastics One, Roanoke, VA, USA). A computer and programming software (Med PC-IV; Med Associates, St Albans, VT, USA) controlled all operant sessions and data collection.

#### 4.2.2. Surgery

A stainless-steel monopolar electrode (MS303/1; Plastics One, Roanoke, VA, USA) was implanted into rats under sodium pentobarbital anesthesia (50 mg/kg, interperitoneally), guided by the atlas of Paxinos and Watson [27]. A cathode (0.25 mm in diameter, coated with polyamide insulation except at the tip) was implanted into the left MFB at the level of the hypothalamus (2.76 mm posterior to the bregma, 1.7 mm lateral to the midsagittal suture, 8.8 mm ventral to the skull), and the anode (0.125 mm in diameter, non-insulated) was wrapped around one screw to serve as the ground. The electrode was anchored to four stainless-steel skull screws (Plastics One, Roanoke, VA, USA), and dental acrylic was used to secure the electrode to the screws and to the skull.

#### 4.2.3. ICSS Procedure

The behavioral procedure for ICSS was similar to that described previously [5]. Initially, rats were trained under a fixed-ratio 1 (FR1) schedule of reinforcement. During each training session, the frequency of stimulation was held constant at 141 Hz, and the intensity was adjusted to maintain consistent responsiveness. Each quarter turn of the wheel delivered a 0.5 s train of square-wave cathodal pulses (0.1 ms pulse duration). During the behavioral session, the house light was illuminated, and stimulation was accompanied by illumination of a stimulus light above the response wheel. Following the delivery of brain stimulation, responses during the 0.5 s time-out phase were recorded, but no consequences were applied. The stimulation current (100–300 μA) was individually adjusted for each rat to identify the lowest value that would sustain a reliable rate of response (at least 30 stimulations per min) during 30 min sessions held on three consecutive days. Once an appropriate stimulation current was identified for each animal, this was delivered throughout the remainder of the study. Each rat was then subjected to brief training at each of a descending series of 15 frequencies (141–28 Hz) (a 1 min trial at each frequency). Each trial consisted of an initial 5 s priming phase during which non-contingent stimulation at the designated frequency was delivered at 1 s intervals, a 50 s response phase during which the responses triggered intracranial stimulation at the designated frequency, and a final 5 s time-out phase during which responses had no scheduled consequences. This 15 min component was repeated three or four times, such that a total daily training session lasted 45 or 60 min. The training sessions continued until the mean daily threshold frequencies varied by less than 10% across three consecutive sessions. Once this criterion was met, behavioral testing began.

In the first experiment, three frequency–rate curves were determined immediately before acupuncture. The first curve was discarded because lever-pressing behavior tended to be unreliable during this 15 min period. The second and third curves were averaged to obtain a baseline frequency–rate curve, and data from the four series of 15 frequencies (passes) were averaged to yield a test frequency–rate curve after acupuncture. After obtaining baselines for each rat on each day, the rats were removed from the operant testing chambers, subjected to acupuncture, and four more 15 min rate-frequency curves obtained (1 h of testing in all). In the second experiment, the conditions were similar to those in the first ICSS experiment. Once the baselines were established using the second and third curves, cocaine (3.2 mg/kg) was intraperitoneally injected immediately before acupuncture. After 30 min, the rats were returned to the operant testing chambers for 60 min to obtain four more rate-frequency functions.

The sequence of cocaine and acupuncture treatments was always performed in the same order and was not counterbalanced. We chose a dose of 3.2 mg/kg cocaine based on our preliminary experiment, in which this dose caused a significant reduction in the ICSS threshold. However, at a higher cocaine dose, ICSS rates were elevated across the entire frequency range in most rats after cocaine injection. Thus, the threshold could not be determined after treatment with 10 mg/kg cocaine. The saline group was used as a control for the cocaine treatment group. The order of acupuncture for each group of rats was arranged according to a within-subject Latin-square design.

### 4.3. Acupuncture Treatment

In the first experiment, to investigate the effects of acupuncture on ICSS reward responses, acupuncture at the bilateral HT7 or SP6 acupoint was applied in unanesthetized rats. For acupuncture treatment, stainless-steel needles (Figure 4A) (0.10 mm diameter, 7 mm length; Dongbang Medical Co., Boryeong-si, Chungcheongnam-do, Korea) were inserted vertically to a depth of 3 mm into the acupoints of rats lightly immobilized by hand for 1 min. The needle was manually rotated during insertion and withdrawal at a frequency of 2 Hz for 2 s. This paradigm was successfully used to produce acupuncture effects in our previous studies [16]. The anatomical location of the acupuncture points stimulated in rats corresponded to the acupoints in humans, as described previously [28]. HT7 is located on the transverse crease of the wrist of the forepaw, radial to the tendon of the flexor carpi ulnaris muscle (Figure 4B). SP6 is located at the medial side of the hind limb, 2 mm above the tip of the medial malleolus and posterior to the medial border of the tibia. The SP6 point was used as a control acupuncture point to control for nonspecific actions due to mechanical stimulation at a certain area. The HT7 acupuncture point on the heart channel has been used to treat mental and psychiatric disorders in oriental medicine clinics. The SP6 acupuncture point is effective in relieving pain associated with primary dysmenorrhea. In the control group, acupuncture was not applied; rather, an assistant lightly restrained the rat for 1 min in the same manner as in the acupuncture treatment but without needle insertion. Before the application of acupuncture treatment, all rats were handled daily for 5 min over 5 consecutive days to facilitate handling and reduce stress.

### 4.4. Data Analysis

The reinforcement rate was defined as the mean of the responses during each frequency trial after acupuncture. The ICSS threshold and the half-maximal frequency (M_50_) were determined from the frequency–rate function by linear interpolation of points between the 20% and 80% maximal responses, and the first points below 20% and above 80% of the maximal responses, after transformation of the raw data to maximal response percentages (Prism; GraphPad Software Inc., San Diego, CA, USA). The percentage baseline stimulations also reflect ICSS performance. To obtain these data, the total number of stimulations per component across all frequencies was determined, and the average number of stimulations per test component expressed as a percentage of the average number of stimulations per baseline component during each session. These behavior-dependent measures were chosen because the ICSS threshold and M_50_ are used to assess changes in the function of the brain reward system after various manipulations in ICSS experiments. The percentage baseline stimulation and maximal response rates also reflect changes in motor performance [5]. Data were analyzed using a one-way ANOVA and Fischer’s least significant difference (LSD) post-hoc test. The time course of acupuncture effects on ICSS thresholds and maximal response rates was analyzed using two-way repeated-measures ANOVA. The significance criterion was set to *p* < 0.05.

### 4.5. Histology

At the end of the experiments, the rats were given an overdose of sodium pentobarbital (80 mg/kg, intraperitoneally) and perfused intracardially with saline, followed by formalin buffer, to confirm the locations of the electrodes.

## Figures and Tables

**Figure 1 ijms-22-07519-f001:**
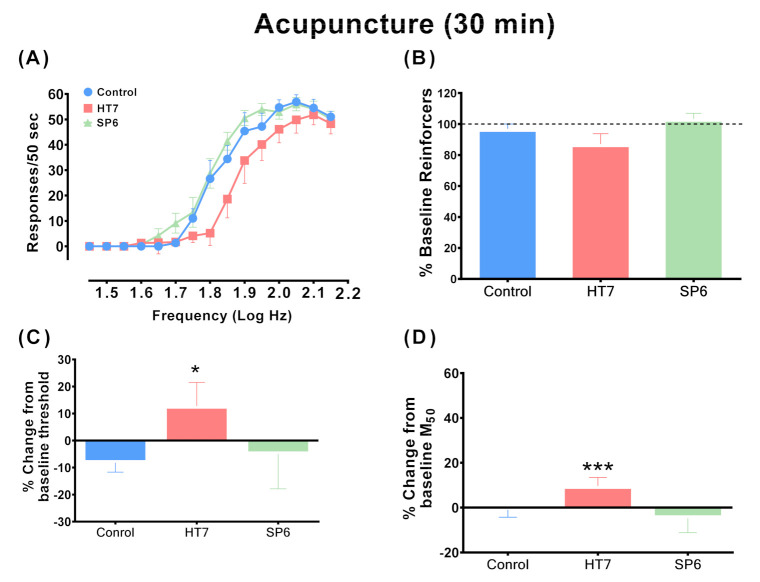
The effects of acupuncture on intracranial self-stimulation during 30 min test sessions in drug-naïve rats. (**A**) Response rates on wheel turning (per 50 s) as a function of stimulation frequency after acupuncture treatment. *Abscissae:* Frequencies of electrical brain stimulation in log Hz. *Ordinates:* Mean responses during each frequency trial after acupuncture. The percentage baseline reinforcer values (**B**), intracranial self-stimulation (ICSS) reward thresholds (**C**) and M_50_ values (**D**) after acupuncture are also shown. The reinforcer percentages are expressed as percentages of baseline total stimulations across all frequencies of brain stimulation per test. Each point represents a mean ± SEM change from baseline. Control group, *n* = 7; HT7 group, *n* = 6; SP6 group, *n* = 7; * *p* < 0.05, *** *p* < 0.001 compared to the control group; post-hoc LSD test.

**Figure 2 ijms-22-07519-f002:**
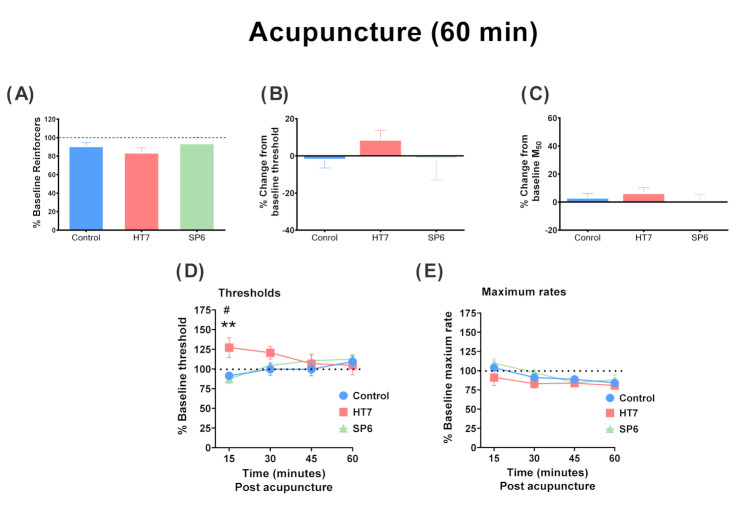
Effects of acupuncture on intracranial self-stimulation during the 1 h test sessions in drug-naïve rats. The percentage baseline reinforcer values (**A**), intracranial self-stimulation (ICSS) reward thresholds (**B**) and M_50_ values (**C**) after acupuncture are also shown. The reinforcer percentages are expressed as percentages of baseline total stimulations across all frequencies of brain stimulation per test. Time (pass)-dependent effects of acupuncture on ICSS thresholds (**D**) and maximum rates (**E**). Each point represents a mean ± SEM change from baseline. Control group, *n* = 7; HT7 group, *n* = 6; SP6 group, *n* = 7, ** *p* < 0.01 compared to the control group; post-hoc LSD test. # *p* < 0.05 compared to the SP6 group.

**Figure 3 ijms-22-07519-f003:**
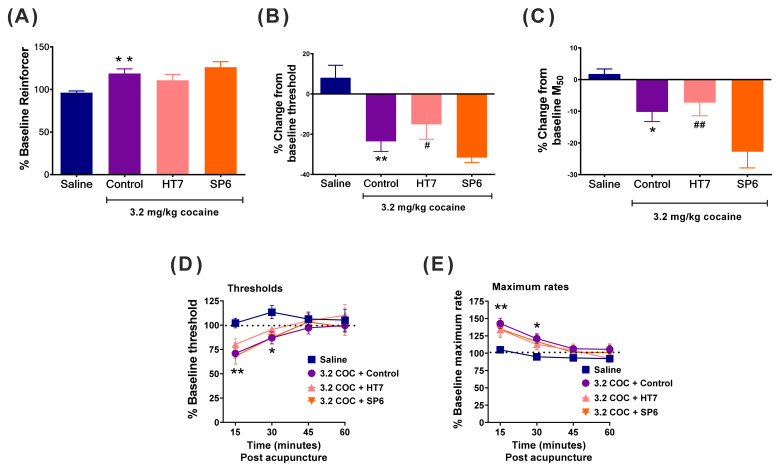
Effects of acupuncture on the cocaine (3.2 mg/kg)-induced shifts in the percentage baseline reinforcer values (**A**), the intracranial self-stimulation (ICSS) reward thresholds (**B**) and the M_50_ values (**C**). Data are expressed as the means of those derived during 1 h test sessions. Time (pass)-dependent effects of combined treatment with acupuncture and cocaine on ICSS thresholds (**D**) and maximum rates (**E**). Each point represents a mean ± SEM change from baseline. Saline group; *n* = 7; control group, *n* = 8; HT7 group, *n* = 7; SP6 group, *n* = 7; * *p* < 0.05, ** *p* < 0.01 compared to the saline group; post-hoc LSD test. # *p* < 0.05, ## *p* < 0.01 compared to the SP6 group.

**Figure 4 ijms-22-07519-f004:**
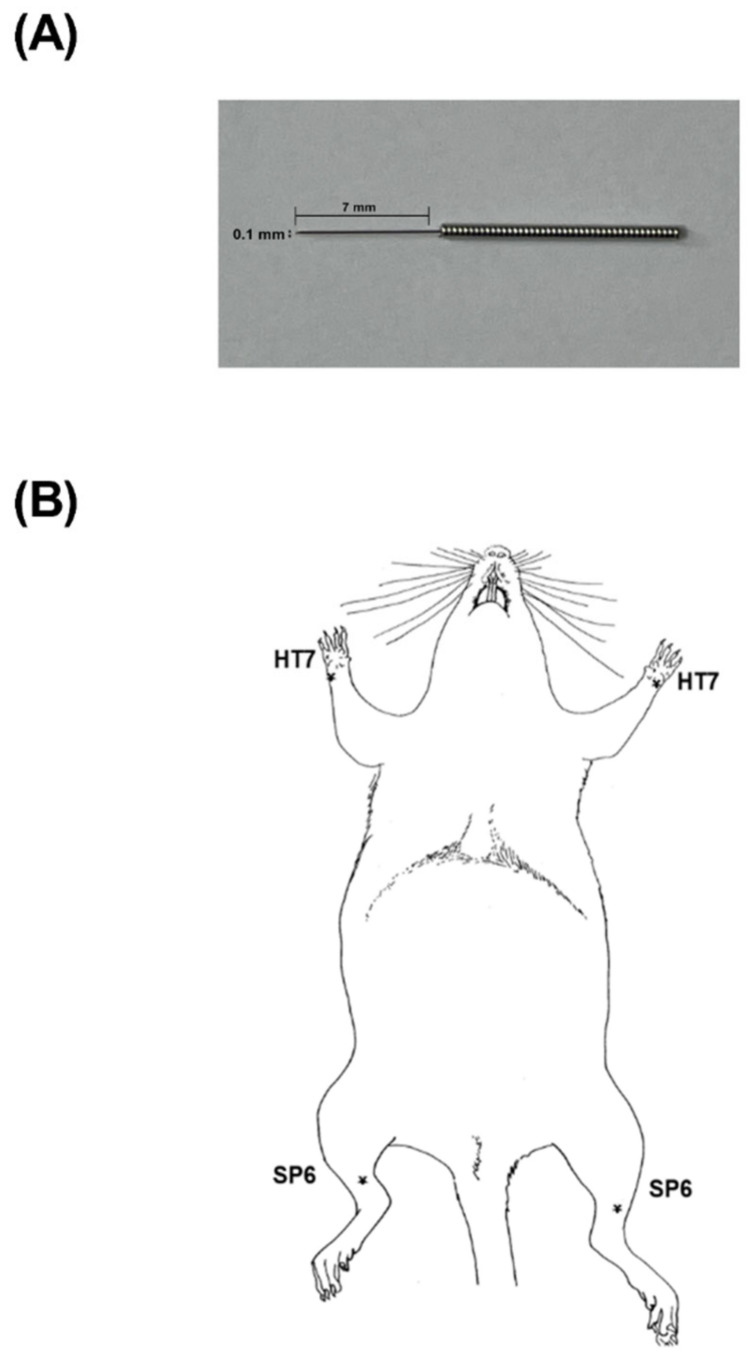
A shape of acupuncture needle (**A**) and the anatomical location of acupuncture points stimulated in rats (**B**).

**Table 1 ijms-22-07519-t001:** Mean (± SEM) ICSS reward thresholds (log Hz) and M_50_ values during baseline sessions before acupuncture treatment. The baseline values before acupuncture did not differ significantly among the groups.

Thresholds and M_50_ Value Obtained during the Baseline before Acupuncture Treatment			
Experiment	Group	Thresholds ± SEM (log Hz)	M_50_ ± SEM (log Hz)
Experiment 1	Control (*n* = 7)	1.73 ± 0.6	1.82 ± 0.5
	HT7 (*n* = 6)	1.75 ± 0.7	1.86 ± 0.6
	SP6 (*n* = 7)	1.7 ± 0.4	1.82 ± 0.6
Experiment 2	Control (*n* = 8)	1.68 ± 0.5	1.78 ± 0.6
	Saline (*n* = 7)	1.64 ± 0.3	1.79 ± 0.5
	HT7 (*n* = 7)	1.67 ± 0.6	1.79 ± 0.7
	SP6 (*n* = 7)	1.69 ± 0.4	1.77 ± 0.6

## Data Availability

Not applicable.

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
