# Peer review of "Acupuncture Modulates Intracranial Self-Stimulation of the Medial Forebrain Bundle in Rats"

_ijms, 2021, doi:10.3390/ijms22147519_

Round 1

Reviewer 1 Report

I am satisfied with changes that were made by the Authors. I think that now the paper is acceptable.

Author Response

This study has been improved considerably by your expertise. We appreciate the time and efforts of the reviewer. 

Reviewer 2 Report

The aim of the study was to investigate the effect of acupuncture on the brain's reward function using the ICSS paradigm in an animal model. The authors showed that HT7 acupuncture point stimulation induced a right-shift in the frequency curve and raised ICSS thresholds, but did not affect the cocaine-induced threshold lowering effect. Based on the results of the study, the authors concluded that HT7 acupuncture points efficiently regulate ICSS thresholds only in the medial forebrain bundle of drug naïve rats.
The article was written concisely and clearly. The results of the study bring new information about the effects of stimulation of acupuncture points.

Author Response

This study has been improved considerably by your expertise. Thank you for the time and efforts.

This manuscript is a resubmission of an earlier submission. The following is a list of the peer review reports and author responses from that submission.

Round 1

Reviewer 1 Report

The manuscript herein presented has minimal scientific interest to the International Journal of Molecular Sciences. It describes a study where intracranial self-stimulation (ICSS) by acupuncture was used to evaluate whether HT stimulation regulates the brain reward function of rats.

The paper is confused and hard to understand its content. At the experimental level, the total number of animals is not described. In every result showed, it is written n=7-8. This is not acceptable. An in vivo study should be well thought, and the number of animals clearly described. The results are not well organized. In terms of language, there are English mistakes throughout the document, so a full correction should be performed. For example, “drugs of abuse”.

Reviewer 2 Report

The communication should be accepted after a minor revision that should focus on following points:

  1. The Authors should make comments about the data shown in Table 1. How is this data related to data presented in Figures 1-3 ?
  2. What is the time after acupuncture for experiments performed on cocaine-treated rats ? (Figure 3)
  3. Did the Authors estimate statistical significance between the data obatined for HT7 and SP6 acupuncture in Figure 3 ? It seems that HT7 acupuncture can slightly diminish the cocaine-induced shifts, whereas SP6 acupuncture can slightly increase it. The effects exerted by different acupuncture methods are likely to be significantly different from each other. Can the Authors make some comments about it ?